# OpenReview forum: "Flow of Reasoning: Training LLMs for Divergent Reasoning with Minimal Examples"
_ICML.cc/2025/Conference — ICML 2025 poster_

### Official Review · Reviewer_PdTd · 2025-03-10

**Overall Recommendation:** 3

**Summary:**

This paper introduces "Flow of Reasoning" (FOR), a novel, data-efficient method for finetuning large language models (LLMs) to achieve divergent reasoning, i.e., generating multiple, diverse, and valid solutions to multi-step reasoning problems. FOR formulates LLM reasoning as a Markovian flow on a directed acyclic graph and adapts principles from Generative Flow Networks (GFlowNets) to train the LLM to sample reasoning paths with probabilities proportional to the problem's reward. The authors show, across six challenging reasoning benchmarks (including BlocksWorld, Game24, Rubik's Cube, 1D-ARC, GSM8k, and ProntoQA), that FOR, with minimal training examples (around 15), significantly outperforms various baselines including supervised finetuning, reward-maximization reinforcement learning, and prompting-based methods, in terms of both solution accuracy and diversity. The method leverages efficient exploration techniques including local search and a prioritized replay buffer to improve the training process.


## update after rebuttal
As discussed during the rebuttal period, I think some minor issues still exist, and the authors need to add some details to the paper. I would like to maintain my score.

**Claims And Evidence:**

1. The paper focuses on 6 main benchmark tasks. It also performs some testing of OOD transfer from smaller to large problems. While testing these benchmarks, how representative are they of all multi-step reasoning problems? There is limited discussion on the types of reasoning these benchmarks don't cover. For example, are there types of reasoning (e.g. causal, counterfactual) for which this approach is expected not to be a good choice?
2. Computational Cost of GFlowNets vs. Alternatives: The introduction and related work sections highlight the potential computational cost of search-based inference methods (ToT, RAP). While FOR amortizes inference cost into training, the paper acknowledges in Appendix C.3 that FOR's training time is significantly higher than SFT and even PPO (Table 7: FOR 6833s, SFT 196s, PPO 1740s). This crucial point needs to be discussed more prominently in the main text, not just the appendix. The trade-off between inference speed and training cost should be explicitly addressed. The claim of efficiency is somewhat misleading without this context. The paper argues for amortized inference, but a user concerned with overall computational cost (including training) might still prefer a slower inference method with much faster training.

**Essential References Not Discussed:**

N/A

**Experimental Designs Or Analyses:**

1. The paper mentions that search-based methods (ToT, RAP) and O1-series are evaluated with limited runs due to time and budget constraints. This is a significant limitation, as it can lead to an inaccurate assessment of their performance, especially regarding diversity and creativity, which are inherently stochastic. The paper should either conduct more runs (ideally) or clearly acknowledge the potential for underestimating these baselines' capabilities.
2. The ablation study analyzes the impact of removing each component individually. It would be helpful to also discuss potential interactions between these components. For example, is the replay buffer more important when local search is not used? A more comprehensive ablation study, though potentially computationally expensive, would explore removing combinations of components.

**Methods And Evaluation Criteria:**

1. The paper uses a diversity metric based on manually annotating 50 test examples to evaluate semantic differences between solutions in GSM8K. While this is a reasonable approach, manual annotation can be subjective, and 50 examples might be insufficient for a robust evaluation of diversity in a task as open-ended as mathematical reasoning. The paper should acknowledge the limitations of this approach and, if possible, explore alternative or supplementary diversity metrics that are less reliant on manual annotation, or increase sample size. For instance, using different paraphrases, could be an automated way to detect different solution paths.

**Other Comments Or Suggestions:**

N/A

**Other Strengths And Weaknesses:**

1. While applying GFlowNets to LLM reasoning is novel in this specific formulation, the core techniques (trajectory balance, local search, replay buffer) are adapted from existing GFlowNet literature.
2. The paper acknowledges Takase et al. (2024) as concurrent work but states it is limited to math problems. A more thorough comparison, even if brief, would be valuable. Are there other concurrent works on applying GFlowNets or similar diversity-seeking methods to LLMs that should be acknowledged and differentiated?
3. The algorithm description could be more precise. For example, the "Sample from training dataset" step needs more detail. How are examples selected? Randomly? With specific criteria? The update of the replay buffer D also needs clarification. Are all sampled trajectories added, or only those exceeding a certain reward threshold?

**Questions For Authors:**

1. The paper acknowledges in the appendix that FOR's training time is significantly higher than SFT and PPO. Could you discuss this trade-off between inference speed and training cost more prominently in the main text? In what scenarios would the increased training cost of FOR be justified by the improved inference performance?
2. Have you explored any methods to reduce the training cost of FOR, such as more efficient exploration strategies or approximations of the GFlowNet objectives?
3. How does the performance of FOR change with different base LLMs?

**Relation To Broader Scientific Literature:**

The paper leverages GFlowNets, extending their application beyond established domains like molecule generation to multi-step reasoning, similar to but distinct from concurrent work like GFN-CoT by focusing on reasoning steps rather than token-level generation. The work incorporates ideas from reinforcement learning exploration, adapting methods like local search and prioritized replay buffers to the GFlowNet framework for LLMs. Finally, it positions itself within the broader context of LLM prompting and fine-tuning, presenting an alternative approach that attempts to address data requirements of supervised methods and the lack of diversity focus in reward-maximization reinforcement learning.

**Theoretical Claims:**

N/A

---

> ### Author Rebuttal · Authors · 2025-04-01
>
> Thanks for your insightful comments and suggestions.
> - Q1. How representative are the benchmarks? Are there reasoning types where it fails?
>
> A1: Some tasks like GSM8K and Game24, are popular LLM benchmarks, while others, like BlocksWorld and 1D-ARC, are known to be challenging for LLMs. FoR remains a general framework for multi-step reasoning: whenever a problem can be decomposed into intermediate states and has a well-defined reward, we expect FoR to be applicable.
>
> - Q2: Tradeoff between amortized inference vs. slower inference with faster training. When training cost of FoR is justified by its better inference performance?
>
> A2: Unlike methods like ToT/RAP without diversity, FoR enables efficient amortized inference with diversity. Though training time is long, it eliminates abundant inference costs in ToT/RAP, keeping the overall cost low when the number of tasks needed to be solved is numerous. Once trained, it avoids repeated, expensive searches. In such cases, the initial training cost is largely offset by the benefits of efficient inference. Compared to SFT/PPO, FoR has higher training costs but is highly data-efficient, needing only ~15 examples, reducing data collection costs.
>
> - Q3: Diversity metric (human annotation & automatic)
>
> A3: Following your advice, we use a paraphrase-based method [1] with GPT-4o to automatically evaluate solution diversity on GSM8K. Please see detailed results in A2 to reviewer C9xd due to page limit. The results show that FoR’s diversity is still higher than baselines by using GPT4o, although it consistently overestimates.
>
> - Q4: Evaluation of diversity for search-based methods and O1-series.
>
> A4: We ran ToT-DFS multiple times on the Blocksworld task with the same setup as in the paper. The results are shown below:
>
> ||2-step|4-step||6-step||
> |-|-|-|-|-|-|
> |Method|Acc.(%)|Acc.(%)|Diversity|Acc.(%)|Diversity|
> |ToT-DFS|40.0|42.9|1.0|31.3|1.1|
> |FoR|100|98.4|1.3|78.4|1.3|
>
> Search methods (ToT/RAP) show limited diversity despite multiple runs. Table 1 in the paper shows ToT/RAP needs 4-40× FoR's inference time per sample, making equal-sample comparison time-unfair. We'd like to clarify that Table 1 shows that O1-mini achieves decent accuracy but limited diversity across multiple runs.
>
> - Q5: Replay buffer is more important without local search?
>
> A5: We add an additional ablation study by removing the local search to assess the impact of the other components in FoR. See the results below:
>
> |Method|4-step Acc.(%)|4-step Div.|6-step Acc.(%)|6-step Div.|
> |-|-|-|-|-|
> |FoR w/o local search|89.7|1.2|53.9|1.3|
> |w/o replaybuffer|78.6|1.1|34.3|1.2|
> |w/o ϵ-sampling|83.3|1.1|49.5|1.1|
> |w/o augmented reward|71.4|1.0|30.3|1.2|
>
> Table 5 in the paper and the above results show that removing the replay buffer causes a smaller performance drop in FoR with local search (4-6%) than without it (11-19%), indicating that the replay buffer is more critical when the local search is removed.
>
> - Q6: While applying GFlowNets to LLM reasoning is novel, core techniques are adapted from existing GFlowNet literature.
>
> A6: Thanks for recognizing our novel formulation, which allows us to adapt existing approaches to the new multi-step LLM reasoning domain. This ability to reuse and generalize, rather than reinvent everything from scratch, is a key advantage of our work.
>
> - Q7: Related works.
>
> A7: We'll add more discussion on Takase et al. (2024) in the revision. Their work generates diverse math solutions via token-level diversity, similar to Hu et al. (2024) (mentioned in our Introduction, Lines 95–96). Oh et al. (2024) [2] follow these lines and apply GFlowNets to LLM preference optimization. In contrast, FoR targets broader reasoning tasks beyond math or preference learning, modeling thought process structures with GFlowNets.
>
> - Q8: Details on training data sampling and replay buffer update mechanism.
>
> A8: We used random sampling from the full training set. For the replay buffer, we set the capacity to store 50 trajectories, new trajectories will replace the lowest-reward ones when the buffer is full. These details will be added to Appendix D.
>
> - Q9: More methods (e.g. efficient exploration) to reduce the training cost.
>
> A9: We use (1) local search and (2) ϵ-sampling (Sec. 4.7) to encourage efficient exploration. In Game24, we use offline data to reduce model's exploration costs. Other methods to accelerate training are a promising future direction.
>
> - Q10: Different base LLMs.
>
> A10: We test LLama-3-8B, Qwen-2.5-7B, and InternLM2.5-7B-Chat on Blocksworld (one task due to rebuttal constraints), and FoR consistently outperforms baselines (see A5 to reviewer c9xd due to page limit).
>
> References:
>
> [1] Michail et al. "PARAPHRASUS: A Comprehensive Benchmark for Evaluating Paraphrase Detection Models." COLING 2025.
>
> [2] Oh Joon Kwon, et al. GDPO: Learning to Directly Align Language Models with Diversity Using GFlowNets. EMNLP 2024

---

### Official Review · Reviewer_KgwZ · 2025-03-14

**Overall Recommendation:** 3

**Summary:**

The paper introduces a novel fine-tuning method called Flow of Reasoning (FOR) that trains large language models (LLMs) to generate diverse, high-quality multi-step reasoning paths using minimal training examples. The key idea is to formulate the reasoning process as a Markovian flow on a directed acyclic graph (DAG) and to leverage GFlowNet-inspired training objectives. By assigning probabilities to entire reasoning trajectories proportional to an (unnormalized) reward, FOR encourages the discovery of multiple valid solution paths. Extensive experiments on six reasoning tasks demonstrate that FOR not only improves overall accuracy but also significantly enhances the diversity and creativity of the generated solutions compared to standard methods such as supervised fine-tuning (SFT), reward-maximizing reinforcement learning , and various prompting-based approaches.

**Claims And Evidence:**

The claims made in the submission are well-supported by empirical evidence:

1. The superiority of FOR over baselines is demonstrated across six diverse reasoning tasks with consistent improvements in accuracy, diversity, and creativity metrics (Tables 1-6).

2. The ablation studies (Table 5) convincingly show the contribution of each component (local search, augmented rewards, replay buffer, ε-sampling).

3. Data efficiency is well-demonstrated through comparisons with SFT trained on varying amounts of data (Figure 3).

4. The case studies in Figures 5-6 provide qualitative evidence of FOR's ability to discover diverse solution paths.

**Essential References Not Discussed:**

To my knowledge, there are no essential references not discussed.

**Experimental Designs Or Analyses:**

The experimental designs are robust:

* Task Diversity: Evaluations span across several reasoning domains (embodied, mathematical, spatial, abstraction), which reinforces the generality of the approach.
* Baselines and Metrics: The comparisons with both prompting-based methods and various fine-tuning strategies, along with clearly defined evaluation metrics, add rigor to the analysis.

One point for further improvement would be a deeper analysis of how hyperparameters (especially in reward design) affect the outcomes. Additionally, exploring the method’s performance with varying amounts of training data could further highlight its data efficiency.

**Methods And Evaluation Criteria:**

The proposed method’s formulation of multi-step reasoning as a Markovian flow is both novel and well-motivated. Using the trajectory balance constraint to link the flow to rewards is theoretically sound and aligns with recent advances in GFlowNets. The evaluation criteria—accuracy, diversity (semantic differences among correct solutions), creativity (unique solutions), and runtime efficiency—are appropriate for the task at hand and provide a comprehensive picture of the method’s strengths and limitations.

**Other Comments Or Suggestions:**

Suggestions:
* Consider including a more detailed discussion on the sensitivity of the method to different hyperparameter settings, especially in the reward design.
* A discussion on potential computational overhead or scalability challenges when applying the method to larger models or datasets would strengthen the paper.
* Clarifying the limitations and potential failure modes of the method could provide a balanced view of its applicability.
* Try to generalize the current FoR to wider range of challenging reasoning tasks, such as Math and Code.

**Other Strengths And Weaknesses:**

Strengths:

* Novel Formulation: The flow-based perspective and the integration of GFlowNet techniques are both innovative and promising.
* Comprehensive Evaluation: The extensive experimental validation across multiple reasoning tasks is a significant strength.
* Data Efficiency: The method’s ability to work with minimal training data is particularly compelling.

Weaknesses:

* Complexity in Derivations: Some of the theoretical derivations might be challenging for readers not already familiar with GFlowNets.
* Hyperparameter Sensitivity: The reliance on handcrafted reward designs and specific hyperparameters might limit the method’s generalizability without further analysis.
* Scalability: While the approach works well on the tasks presented, its performance when scaling to larger models or more open-ended tasks is not fully explored.

**Questions For Authors:**

Question:

1. How sensitive is FOR to the reward design? Could you provide guidelines for designing rewards for new tasks to help understand the generalizability of your approach?

2. The training cost of FOR is significantly higher than SFT (Table 7). Are there optimizations that could make the approach more computationally efficient without sacrificing performance?

3. How does FOR perform when the number of possible reasoning paths grows very large? Are there scaling limitations when applying this to more complex reasoning tasks?

4. Have you explored combining FOR with other techniques like verifiers or self-consistency approaches? This might address some of the limitations for longer-range reasoning.

5. How did you determine the reward weights (e.g., λ values) for each task? Is there a systematic approach to tuning these hyperparameters?

**Relation To Broader Scientific Literature:**

The key contributions of the paper are related to the current research LRMs, which could potentially be an approach to refine the thinking and exploration process of the current reasoning trajectory.

**Theoretical Claims:**

The paper primarily presents a conceptual framework rather than making formal theoretical claims requiring proofs. The authors adapt existing GFlowNet theory to the domain of reasoning steps in a sound manner.

---

> ### Author Rebuttal · Authors · 2025-04-01
>
> Thanks for recognizing our strong performance across 6 benchmarks, ablation studies, data efficiency, and case studies.
>
> - Q1: Hyperparameters selection and analysis? Guidelines for designing rewards? Handcrafted rewards?
>
> A1: Please refer to Section 4.7 and Figure 3 for the hyperparameter analysis on a small training set (i.e., 15). We search for the best-performing λ values on each task's training set.
>
> We further assess how the intermediate reward weight λ affects test set performance. Results for varying λ values are shown below:
>
> |λ|0||0.5||1||1.5||2||
> |-|-|-|-|-|-|-|-|-|-|-|
> |FoR|Acc.(%)|Div.|Acc.(%)|Div.|Acc.(%)|Div.|Acc.(%)|Div.|Acc.(%)|Div.|
> |4-step|90.5|1.2|92.9|1.2|97.6|1.2|98.5|1.3|95.2|1.3|
> |6-step|47.1|1.2|71.7|1.3|75.8|1.3|78.4|1.3|76.8|1.3|
>
> The results show that performance consistently improves as λ increases up to 1.5, but drops a bit at λ= 2 across 4- to 6-step settings. The influence of λ becomes more pronounced with more steps.
>
> For the guidelines, we use the generalizable common principles: 1. For tasks with likely correct outputs (e.g., GSM8K), a binary success reward suffices. 2. For challenging reasoning tasks (e.g., BlocksWorld, Game24) with sparse rewards, intermediate rewards from prior works (Hao et al., 2023) (e.g., LLM log-likelihood, rule-based) help.
>
> Regarding handcrafted rewards, in GSM8K, we simply use a standard outcome reward (0/1). Additionally, we use the reward model Qwen2.5-Math-PRM-7B, achieving 62.62% accuracy and 1.31 diversity (please see A2 to reviewer c9xd due to page limit). These two reward designs show that FoR does not rely on sophisticated reward design, supporting general applicability.
>
> We will include all these in the revision.
>
> - Q2: Varying amounts of training data.
>
> A2: We ran additional experiments on the 6-step BlocksWorld task using Llama-3-8B with varying training sizes {1, 15, 30, 45, 60}, following the same setup as Section 4.7. The results are shown below:
>
> |Method|1||15||30||45||60||
> |-|-|-|-|-|-|-|-|-|-|-|
> ||Acc.(%)|Div.|Acc.(%)|Div.|Acc.(%)|Div.|Acc.(%)|Div.|Acc.(%)|Div.|
> |SFT|15.0|1.0|40.0|1.0|50.0|1.0|60.0|1.1|70.0|1.0|
> |FoR|45.0|1.3|80.0|1.3|85.0|1.3|90.0|1.3|90.0|1.4|
>
> FoR consistently outperforms SFT across all data sizes—for example, with just one training example, FoR achieves a relative 200% higher accuracy and a relative 30% higher diversity. This highlights FoR’s data efficiency even in low-resource settings.
>
> - Q3: Theoretical derivations.
>
> A3: Please refer to Appendix B for background information on GFlowNets.
>
> For the trajectory balance (TB) constraint, please see Section 3.2. Below is a short derivation of the TB constraint:
>
> Let $P_F(τ) = \prod_{t=1}^{n} P_F(s_t \mid s_{t-1})$ be the forward trajectory distribution, and $P_B(τ) = \prod_{t=1}^{n} P_B(s_{t-1} \mid s_t)$ be the backward trajectory distribution.
>
> The **TB constraint** requires: $P_F(τ) \cdot Z(s_0) = P_B(τ) \cdot R(s_n)$.
>
> This constraint aligns the flow allocated to $τ$ under the forward policy with reward-scaled backward probability, enabling proper credit assignment.
>
> We will add these in the revision.
>
> - Q4: Scalability and open-ended tasks.
>
> A4: For larger models, we run additional experiments with LLaMA-3-70B and Qwen2.5-72B, and FoR consistently achieves better accuracy and diversity (see A5 to reviewer c9xd due to page limit). On the open-ended GSM8K benchmark, FoR also achieves stronger performance, highlighting its scalability.
>
> - Q5: Limitations and potential failure modes.
>
> A5: Please see the Limitations in Appendix H and will move it to the main text. For potential failure modes, FoR, like other on-policy RL methods, may struggle in sparse-reward settings. However, FoR mitigates this by the augmented intermediate rewards (Sec. 4.7), which provide dense guidance, and by leveraging off-policy data (Sec. 4.3 Game24).
>
> - Q6: More reasoning tasks, such as Math and Code.
>
> A6: We’d like to clarify that FoR has already been evaluated on GSM8K, a math benchmark (Sec. 4.6), as well as on ARC-1D (Sec. 4.5), which involves Python program synthesis.
>
> - Q7: Optimizations to be computationally efficient?
>
> A7: We'd like to clarify that although FoR’s training is slower, it is data-efficient (Section 4.7), requiring only ~15 training examples—keeping the overall training cost low. Two potential further optimization directions are: 1. Off-Policy Data: Parallel off-policy training can reduce the computational cost (Appendix H). 2. Intermediate rewards: accurate reward functions (e.g., robust reward models) enable faster convergence.
>
> - Q8: A large number of reasoning paths? Scaling limitations?
>
> A8: FoR performs well with large reasoning spaces. For example, the Game24 task involves ~8,000 distinct reasoning trajectories per sample, showing that FoR could scale to complex reasoning tasks.
>
> - Q9: Other techniques like verifiers?
>
> A9: We use rule-based rewards as verifiers in Rubik’s Cube and 1D-ARC (Sec. 4.4&4.5). Your suggestion is also a promising direction.

---

### Official Review · Reviewer_C9xd · 2025-03-18

**Overall Recommendation:** 4

**Summary:**

The paper introduces Flow of Reasoning (FOR), a method for training Large Language Models (LLMs) to generate diverse, high-quality reasoning paths with minimal training examples. The authors formulate multi-step LLM reasoning as a Markovian flow on a DAG-structured reasoning graph, adapting Generative Flow Networks (GFlowNets) to train LLMs to sample reasoning paths with probabilities proportional to their rewards.
The key innovation is enabling "divergent reasoning" - generating multiple valid solutions to a problem rather than just maximizing rewards for a single solution path. FOR incorporates local search with destroy-and-reconstruction processes to augment training trajectories, and uses both online and offline exploration strategies including replays and ε-sampling.
The method demonstrates superior performance across six challenging reasoning tasks (BlocksWorld, Game24, Rubik's Cube, 1D-ARC, GSM8k, and ProntoQA), outperforming prompting-based methods and fine-tuning approaches in both accuracy and solution diversity with only ~15 training examples.

**Claims And Evidence:**

The claims are well-supported by comprehensive experiments:

1. Performance comparisons with numerous baselines (CoT, ToT, RAP, SFT, PPO, GFN-CoT)
2. Detailed ablation studies demonstrating the contribution of each component
3. Quantitative metrics for accuracy, diversity, and creativity
4. Analysis of data efficiency showing FOR's effectiveness with limited examples
5. Well-designed case studies illustrating how FOR discovers multiple correct solutions

The methodology is sound and the empirical results convincingly demonstrate FOR's advantages.

**Essential References Not Discussed:**

The paper generally covers relevant literature thoroughly.

**Experimental Designs Or Analyses:**

Report performance across multiple runs with standard deviations
Define clear metrics that directly measure their objectives
Include OOD testing to demonstrate generalization capabilities
Provide detailed implementation specifications in the supplementary material

**Methods And Evaluation Criteria:**

The methods and evaluation criteria are appropriate for the problem:

The six reasoning tasks cover diverse domains (embodied, mathematical, spatial, abstraction, etc.)
The metrics (accuracy, diversity, creativity) directly address the paper's goal of divergent reasoning
The baselines represent state-of-the-art methods in both prompting and fine-tuning approaches
The ablation studies isolate the contributions of individual components

The reward designs for each task are thoughtfully crafted to balance accuracy and exploration.

**Other Comments Or Suggestions:**

Nothing critical to comment

**Other Strengths And Weaknesses:**

Strengths:

Novel formulation of LLM reasoning as a Markovian flow problem
Data efficiency (works with ~15 examples)
Comprehensive evaluation across diverse reasoning tasks
Clear ablation studies demonstrating component contributions
Practical runtime comparable to efficient baseline methods

Weaknesses:

The paper doesn't address integration with larger models or more complex real-world tasks
The trajectory diversity metric could be more sophisticated to better capture semantic differences
Limited exploration of other reward formulations that might further enhance diversity
No analysis of whether diversity truly benefits downstream applications

**Questions For Authors:**

How would the method perform with even larger models (70B+)?
Would the gains in diversity increase or decrease relative to baselines?
Could the approach be extended to iterative reasoning refinement settings where solutions are revised based on feedback?
Have you explored whether a meta-learning approach could reduce the need for task-specific reward engineering?

**Relation To Broader Scientific Literature:**

The paper builds on three critical areas:

LLM reasoning: Extends CoT/ToT approaches by enabling diverse reasoning paths
GFlowNets: Adapts them from molecule generation to structured reasoning
Diverse sampling methods: Provides a principled alternative to beam search

Unlike previous applications of GFlowNets with LLMs, FOR implements higher-level modeling at the reasoning step granularity rather than token level, which is key to its success in reasoning tasks.

**Theoretical Claims:**

There doesn't seems to has concrete theoretical claim

---

> ### Author Rebuttal · Authors · 2025-04-01
>
> Thank you for recognizing our innovative Markovian flow approach to multi-step LLM reasoning with GFlowNets, our strong performance across six benchmarks, and our method’s data efficiency.
>
> - Q1: More complex real-world tasks?
>
> A1: We evaluate FoR on six benchmarks that pose significant challenges for current LLMs (e.g., embodied reasoning in BlocksWorld and math reasoning in GSM8K). These tasks capture core difficulties encountered in real-world scenarios. For instance, the BlocksWorld task requires not only spatial reasoning but also practical world knowledge, closely mirroring real-world planning problems.
>
> - Q2 The diversity metric to capture semantic differences?
>
> A2: Thank you for your suggestion. We intentionally use human annotation for GSM8K to ensure the evaluation is accurate and reliable, despite the high cost of it. Here, we run additional experiments using a paraphrase-based method [1] to automatically evaluate the diversity with GPT-4o to measure the number of different solutions for each problem. The results on GSM8K are shown below:
>
> |Method|Acc. (%)|Div. (Human)|Div. (GPT-4o)|
> |-|-|-|-|
> |CoT (2-shot)|45.72|1.12|1.60|
> |SFT (α = 1.0)|52.69|1.13|1.63|
> |FoR|57.39|1.26|1.72|
> |FoR w/ Process Reward Model (PRM)|62.62|1.31|1.77|
>
> Compared to human annotation, GPT-4o overestimates the reasoning diversity, indicating that these automatic metrics currently do NOT provide more reliable evaluation. So, we leave this evaluation method as future work.
>
> - Q3: Reward formulations that enhance diversity?
>
> A3: Thanks for your inspiring comment. We agree that explicitly incorporating diversity into the reward formulation may further enhance solution diversity. We will discuss this in the next version.
>
> - Q4: Analysis of diversity truly benefits downstream applications
>
> A4: Thanks, we'll further highlight this in the revision. We'd like to clarify that diversity indeed benefits downstream tasks by improving accuracy, as mentioned in the Introduction (Lines 101-104). For a detailed analysis, please refer to Section 4.7 (Experiment Discussion) and Appendix F. We find that diversity promotes exploratory behaviors (e.g. in Game 24, it explores a large trajectory space), thereby enhancing the robustness of the model (Fig. 6).
>
> - Q5: Scale with larger models, and diversity gains compare to baselines?
>
> A5: Given the time and resource constraints in our research lab during the rebuttal period, we ran additional experiments to evaluate FoR's scalability and diversity gains using LLaMA-3 (8B & 70B) and Qwen2.5 (7B & 72B) on the BlocksWorld task. Results, shown below, indicate that FoR consistently improves accuracy and diversity with larger models. Compared to CoT baselines, FoR exhibits greater diversity gains as model size increases. This suggests that the gains of FoR become more pronounced as model capacity increases. Even with minimal data (15 examples), FoR yields clear improvements, highlighting its potential for robust performance across different base models. The results are shown below and will be included in the revision.
>
> |Model|4-Step Acc. (%)|4-Step Div.|6-Step Acc. (%)|6-Step Div.|
> |-|-|-|-|-|
> |CoT 5-shot (Llama3-8B)|28.57|1.05|15.82|1.05|
> |CoT 5-shot (Llama3-70B)|45.23|1.05|46.46|1.11|
> |FoR (Llama3-8B)|98.41|1.27|78.44|1.33|
> |FoR (Llama3-70B)|100.00|1.38|87.65|1.40|
> |FoR (Qwen2.5-7B)|100.00|1.24|86.86|1.36|
> |FoR (Qwen2.5-72B)|100.00|1.41|90.13|1.46|
> |FoR (InternLM2.5-7B-Chat)|100.00|1.26|83.83|1.31|
>
> - Q6: Iterative reasoning refinement based on feedback?
>
> A6: Yes, FoR supports iterative refinement. During both training and inference, by incorporating feedback—from the model itself or an additional model—into the state, subsequent state predictions refine solutions iteratively. This aligns with R1/O1's approach, where each step either refines the current state or advances the reasoning. For example, in BlocksWorld, given a current block position, the model predicts an action like 'move blue onto yellow,' and then generates feedback on whether to proceed or refine. Based on the state including this feedback and block positions, the model predicts the next state, iteratively refining its actions or proceeding. While we haven't tested this, iterative refinement based on feedback is a promising future direction.
>
> - Q7: A meta-learning approach to reduce the need for task-specific reward engineering?
>
> A7: While we haven't directly explored meta-learning approaches, we ran an additional experiment on an existing process reward model, Qwen2.5-Math-PRM-7B, for the GSM8K task. Table in A2 indicates that using this reward model with FoR improves performance over the previous 0/1 outcome reward function, achieving a 9% relative increase in accuracy and a 4% relative gain in diversity. This shows a promising step towards reducing the need for task-specific reward engineering.
>
> References:
>
> [1] Michail et al. (2025) "PARAPHRASUS: A Comprehensive Benchmark for Evaluating Paraphrase Detection Models." COLING 2025.

---

### Decision · Program_Chairs · 2025-05-01

**Decision:**

Accept (poster)

**Comment:**

The authors introduce Flow of Reasoning (FOR), a method for training LLMs to generate diverse, high-quality reasoning paths using minimal training examples by modeling multi-step reasoning as a Markovian flow on a DAG-structured graph, leveraging Generative Flow Networks (GFlowNets). It claims superior performance in accuracy and solution diversity across six reasoning tasks compared to baselines like CoT, ToT, and SFT. Strengths include its novel formulation, data efficiency, and comprehensive evaluation across diverse tasks with clear ablation studies. Weaknesses include limited exploration of scalability to larger models or complex real-world tasks, a basic diversity metric that may not fully capture semantic differences, and high training costs (e.g., 6833s vs. 196s for SFT) not prominently discussed in the main text. The submission could benefit from deeper analysis of hyperparameter sensitivity and applicability to other reasoning types like causal or counterfactual.

During the rebuttal, reviewers raised concerns about training cost trade-offs, diversity metric robustness, scalability, and benchmark representativeness. The authors provided additional experiments, including scalability tests with larger models (LLaMA-3-70B, Qwen2.5-72B) showing consistent gains, a paraphrase-based diversity evaluation with GPT-4o (though less reliable than human annotation), and hyperparameter analysis for reward weights. They clarified FOR’s applicability to math and code tasks and detailed replay buffer mechanics. Reviewer C9xd maintained a strong accept (score 4), appreciating the thorough responses. Reviewer KgwZ, initially a weak accept (score 3), upheld their score, satisfied with most clarifications. Reviewer PdTd, also weak accept (score 3), noted minor unresolved issues like training cost discussion but did not change their score. For the acceptance recommendation, I weighed the robust empirical results and novelty heavily, with the authors’ rebuttal addressing scalability and diversity concerns sufficiently. The training cost issue, while valid, is mitigated by data efficiency and amortized inference benefits. This would be a solid contribution to ICML